# Diagnostic and Prognostic Comparison of Immune-Complex-Mediated Membranoproliferative Glomerulonephritis and C3 Glomerulopathy

**DOI:** 10.3390/cells12050712

**Published:** 2023-02-23

**Authors:** Marja Kovala, Minna Seppälä, Anne Räisänen-Sokolowski, Seppo Meri, Eero Honkanen, Kati Kaartinen

**Affiliations:** 1Department of Pathology, University of Helsinki and Helsinki University Hospital, 00290 Helsinki, Finland; 2Department of Nephrology, University of Helsinki and Helsinki University Hospital, 00290 Helsinki, Finland; 3Translational Immunology Research Program TRIMM, Department of Bacteriology and Immunology, University of Helsinki and Helsinki University Hospital, 00290 Helsinki, Finland

**Keywords:** C3 glomerulopathy, immune-complex-mediated membranoproliferative glomerulonephritis, complement activation, MPGN

## Abstract

Membranoproliferative glomerulonephritis (MPGN) is subdivided into immune-complex-mediated glomerulonephritis (IC-MPGN) and C3 glomerulopathy (C3G). Classically, MPGN has a membranoproliferative-type pattern, but other morphologies have also been described depending on the time course and phase of the disease. Our aim was to explore whether the two diseases are truly different, or merely represent the same disease process. All 60 eligible adult MPGN patients diagnosed between 2006 and 2017 in the Helsinki University Hospital district, Finland, were reviewed retrospectively and asked for a follow-up outpatient visit for extensive laboratory analyses. Thirty-seven (62%) had IC-MPGN and 23 (38%) C3G (including one patient with dense deposit disease, DDD). EGFR was below normal (≤60 mL/min/1.73 m^2^) in 67% of the entire study population, 58% had nephrotic range proteinuria, and a significant proportion had paraproteins in their serum or urine. A classical MPGN-type pattern was seen in only 34% of the whole study population and histological features were similarly distributed. Treatments at baseline or during follow-up did not differ between the groups, nor were there significant differences observed in complement activity or component levels at the follow-up visit. The risk of end-stage kidney disease and survival probability were similar in the groups. IC-MPGN and C3G have surprisingly similar characteristics, kidney and overall survival, which suggests that the current subdivision of MPGN does not add substantial clinical value to the assessment of renal prognosis. The high proportion of paraproteins in patient sera or in urine suggests their involvement in disease development.

## 1. Introduction

Membranoproliferative glomerulonephritis (MPGN) is a pattern seen in light microscopy (LM) in approximately 7–10% of all biopsy-confirmed glomerulonephritis cases [1]. Typical hallmarks are endocapillary and mesangial hypercellularity, mesangial matrix expansion, and formation of capillary double contours resulting in a lobulated morphology [2]. However, these changes can vary from minimal to mesangial, endocapillary proliferative, exudative, crescentic, and sclerosing patterns [3], possibly portraying different time points of injury. Based on immunofluorescence microscopy staining (IF), MPGN is divided into immune-complex-mediated (IC-MPGN) and complement-mediated MGPN (C-MPGN), also known as C3 glomerulopathy (C3G) [4]. In addition, C3G is divided into dense deposit disease (DDD) and C3 glomerulonephritis (C3GN), based on the presence or absence of intramembranous dense deposits on electron microscopy (EM), respectively [4]. EM is also necessary to differentiate organized deposits from MPGN-type deposits [5].

In IC-MPGN, IF reveals immunoglobulins (Igs) and C3 deposits, mainly due to activation of the classical pathway [4], while C3G has bright, dominant, C3 staining at least twice the intensity of any immune reactant, primarily due to abnormal activation of the alternative pathway [6,7,8]. The etiology of IC-MPGN includes autoimmune diseases, chronic infections, malignant diseases, and paraproteinemias [1], such as monoclonal gammopathy of unknown significance (MGUS) or renal significance (MGRS) [9]. An underlying abnormality is lacking in some 30% of cases [10]. In C3G, the alternative pathway is thought to be dysregulated due to gain-of-function or loss-of-function mutations in complement proteins, autoantibodies, or inhibitors against complement regulators [6,11]. However, a subset of patients has no identifiable cause [4,12,13,14]. A triggering second hit is often required to manifest the disease [15,16,17,18].

The clinical presentation of MPGN varies from asymptomatic hematuria and proteinuria to nephrotic or nephritic syndrome, or even rapidly progressive glomerulonephritis. MPGN has a progressive nature and often recurs in kidney transplants [4,19]. Treatment depends on the etiology, but supportive therapy aiming at reducing proteinuria and controlling blood pressure is recommended for all [20,21]. For more severe cases, immunosuppressive therapy, plasma infusions or plasma exchange may be beneficial, but the results vary [1,20,21,22,23]. Some C3G patients may profit from complement C5-inhibitor therapy [21,22,24,25,26,27,28,29]. Several other drugs blocking different sites of the complement cascade have been developed, and are expected to achieve marketing authorization in the near future [11].

Although the information on the pathophysiology of MPGN has greatly advanced in the past 10 years, the diagnosis can still be challenging. The diagnostic accuracy of IF may necessitate proteinase K-processed paraffin-embedded IF (PIF) analysis to show masked immunoglobulins. However, the distinction between C3GN and DDD can sometimes be problematic. In fact, the diagnostic category between IC-MPGN and C3G may even change upon analysis of subsequent biopsies [6,10,30]. The distinction between classical and alternative pathway complement activation is not always straightforward, since the two pathways are interdependent. Alternative pathway activation is common and can also occur in IC-MPGN [31,32]. Diagnostic challenges also exist between postinfectious glomerulonephritis (PIGN) and C3G, as dominant C3 deposition with immunoglobulins can be observed in both entities [11,15,33]. In PIGN, the disease process has been thought to be immune complex-mediated and self-limiting [33]. If the IF result is negative, thrombotic microangiopathy (TMA) should be considered [1,4]. It is postulated that complement activation occurs on cell surfaces in TMA as opposed to mostly fluid-phase activation in MPGN [1,34]. Thus, there is still a great need for further understanding of the mechanisms and diagnostic differences between the various forms of MPGN.

The aim of this study was to investigate the prognostic histological and clinical factors between adult IC-MPGN and C3G patients and to investigate their complement system, both at diagnosis and at the follow-up visit.

## 2. Materials and Methods

### 2.1. Patient Population and Data Retrieval

This single-center study was carried out at Helsinki University Hospital district, Finland, covering roughly 1.7 million inhabitants (31% of the total population). Patients were identified from records in the Department of Pathology. All eligible adults who had received a non-transplanted (native) or transplant biopsy-confirmed diagnosis of MPGN, C3G, TMA, or PIGN from 2006 to 2017 were included. Only the 1st biopsy within the time frame of interest was included (index biopsy), including eight patients who had a diagnostic biopsy taken before 2006. Clinical information was recorded from electronic medical records from the initial diagnosis until the end of 2019, the last visit, death, or when lost to follow-up, whichever occurred latest. The study flow is depicted in Figure 1.

Exclusion criteria were other clear-cut diagnoses merely resulting in an MPGN-type pattern (such as systemic lupus erythematosus or IgA nephropathy), or an inadequate biopsy sample. Other secondary IC-MPGN patients were included. For TMA biopsies, transplant glomerulopathy and antibody-mediated rejection resulting in an MPGN-type pattern were excluded (see the full list of exclusion diagnoses in Appendix A). The eligibility of problematic cases was judged by the research team.

Study was reviewed by the Helsinki University Hospital’s Ethical Committee (HUS/2520/2018) and a research permit (HUS/459/2018) was granted. The study was conducted according to the Declaration of Helsinki. All patients evaluated on the follow-up visit signed a declaration of informed consent.

### 2.2. Laboratory Analyses

Laboratory analyses were performed in the accredited Helsinki University Hospital laboratory using standardized laboratory methods. Hemoglobin was measured from plasma photometrically. The normal ranges for males and females were 134–167 g/L and 117–155 g/L, respectively. Creatinine (upper normal value 100 μmol/L for males and 90 μmol/L for females) and low-density lipoprotein (LDL) (reference values according to Nordic Reference Interval Project [35]) values were analyzed photometrically and enzymatically, respectively. C-reactive protein (CRP) levels were studied photometrically (normal range <4 mg/L). Plasma albumin analysis was carried out photometrically with bromocresol purple reaction, for which the reference values used are from the Nordic Reference Interval Project [35]. Antibodies to extractable nuclear antigens (ENA) were analyzed using a Phadia 250 instrument, and an accredited fluoroenzyme-immunological (FEIA) method with a two-phase test protocol at the HUSLAB Laboratory, Helsinki, Finland. First, the sample was screened for existing antibodies using a known mix of ENA-proteins as an antigen. If the sample was screened positive in the first phase, it was analyzed further against eight ENA antigens (S-JoAb, S-RNP70Ab, S-Scl70Ab, S-SentBAb, S-SSAAb, S-SSBAb, S-SmAb (upper normal limit: 7 U/mL, slightly elevated: 7–10 U/mL and elevated: >10 U/mL), and S-RNPAb (normal: <5 U/mL, slightly elevated: 5–10 U/mL and elevated: >10 U/mL). The type of commercial kit used varied according to the year in which the test was performed.

Estimated glomerular filtration rate (eGFR) was calculated according to the Chronic Kidney Disease Epidemiology Collaboration equation (CKD-EPI), for which the cut-off for lower limit of normal was 60 mL/min/1.73 m^2^. Urine albumin was analyzed photometrically and immunochemically and the upper limit of normal for urine albumin/creatinine ratio was 3.0 mg/mmol. Daily urine protein excretion was analyzed photometrically and with a benzethonium chloride reaction and the upper limit of normal was 100 mg. Microscopic hematuria was analyzed with automatic phase contrast microscopy, and its normal value for erythrocytes was <10 E6/L.

Serum-free light chains were analyzed photometrically and immunochemically using Freelite reagents from The Binding Site, Birmingham, UK. Serum and urine paraproteins were analyzed using immunofixation after agarose gel electrophoresis. Other analyses were performed using routine laboratory methods.

### 2.3. Kidney Biopsies

LM, IF and EM analyses of index biopsies were re-evaluated by pathologists according to the MPGN classification [36], diagnostic criteria of C3G [10], 2018 Banff classification [37] and Definition of Glomerular Lesion by the Renal Pathology Society [38], where appropriate. The overall biopsy morphology was divided into minimal change (morphology nearly normal), MPGN, crescentic (if any crescents were visible), mesangial proliferative (if mesangial matrix expansion and cellularity were seen without double contours), and exudative forms (if glomerular granulocytes were visible) [3,38]. Index biopsy was defined as the first biopsy acquired during the period of interest (2006–2017). Initial diagnostic biopsies procured earlier than 2006 were not available for re-evaluation.

IF brightness was evaluated for IgG, IgM, IgA, C3, C1q, kappa, lambda, and fibrinogen as 0–4, where brightness is negative (0), trace (1), mild (2), moderate (3) or strong (4), respectively. If a frozen section IF was unavailable or masked immunoglobulins were suspected, PIF was performed and evaluated similarly. EM was performed for 41 cases (68%), from which sufficient material was available for the analysis. This included one DDD patient.

IC-MPGN was defined when mesangial and/or capillary immunoglobulins and C3 and/or C1q on IF were detected [4] and C3G was defined when staining was C3 dominant and at least twice brighter than any other reactant on IF [7]. PIGN was defined when a previously diagnosed infection had resulted in kidney injury that resolved within 6 months from the onset of symptoms. TMA was defined when a TMA- or MPGN-type of injury was seen in LM and no (or only a few) immunoglobulins and/or C3/C1q were detected on IF.

### 2.4. Clinical Data

Clinical information was documented from the electronic medical records at diagnosis and during follow-up until the preselected endpoint. Follow-up started from the diagnostic biopsy and ended at the end of 2019 or when the patient moved out of the hospital district, was lost to follow-up, attended the last outpatient visit, or died, whichever occurred latest. All eligible patients were invited for an outpatient follow-up visit, where laboratory tests, differential diagnostics, and complement testing were completed. Patients who had a severe psychiatric illness or severe dementia were excluded from the appointment. Data from their electronic medical records were, however, evaluated.

The time of diagnostic biopsy was set as the baseline. Kidney function was assessed by using both serum creatinine and eGFR. Progressive disease was defined as at least 50% reduction in eGFR from baseline and plasma creatinine level exceeding the upper limit of normal at the last follow-up and/or by kidney failure leading to kidney replacement therapy.

### 2.5. Complement Analyses

Complement analyses were performed for patients attending the outpatient visit. C3 and C4 levels were measured by nephelometry. Anti-factor H-antibodies, anti-C3b-antibodies, and anti-factor B antibodies were quantified using an ELISA assay according to the Helsinki protocol [39]. Human sera without or with autoantibodies were included as negative and positive controls. Nunc Maxisorp plates were coated with 100 μL portions of factor H (Complement Technologies, Tyler, Texas, USA), purified factor C3b, or purified factor B to quantify anti-factor H, anti-C3b, and anti-factor B antibodies, respectively. The plates were then left overnight at 4 °C, whereafter they were washed with phosphate-buffered saline (PBS) containing 0.05% Tween 20 and blocked with 200 μL of the same buffer at ambient temperature for 2 h. Samples were analyzed in duplicate. Serum samples diluted 1/20 were added in 80 μL portions and incubated for 2 h at 37 °C. After washing, HRP-conjugated (horse radish peroxidase, Dako, Glostrup, Denmark) secondary antibodies diluted at 1:2000 in PBS were added and incubated again at 37 °C for 1 h. The plates were washed with PBS and o-phenylenediamine dihydrochloride (OPD) substrate was added. The reaction was stopped at 120 μL of 0.5 M H_2_SO_4_ and a spectrophotometer was used to measure the optical density of samples at a wavelength of 492 nm. C3 nephritic factor (C3Nef) was analyzed by in-house immunofixation electrophoresis, which involved examining the ability of the patient’s serum to activate the alternative pathway of complement in normal serum in the presence of magnesium ethylene glycol tetra-acetic acid (MgEGTA).

Factor H and Factor H-related proteins were analyzed by immunoblotting and compared to normal human serum controls. Appropriately diluted samples were added in 1:100 and 1:300 dilutions into 4–12% SDS-PAGE gradient gels (Thermo Fisher Scientific, Waltham, MA, USA). The samples were then run for 45 min at 165 V in 1× MES buffer, whereafter they were transferred onto a filter membrane. Non-specific binding sites were blocked for 1 h at room temperature with 1 mL of 5% non-fat dry milk prepared in 1× PBS and 0.05% Tween 20. Later, primary goat anti-factor H-antibody was added into the solution and incubated overnight at 4 °C. The membranes were then washed at room temperature for 1 h with 15–20 min changing intervals of 2–3 mL 1× PBS and 0.05% Tween 20. Secondary rabbit-anti-goat HRP-conjugated antibody was added with 5% non-fat dry milk and incubated at room temperature for 1 h. The washing with PBS changing intervals was then repeated at room temperature for 1 h. The bands were visualized with an in-house protocol of electrochemiluminescence recipe for 1 min at room temperature. Films were developed at different exposure times, after which the results were interpreted visually. If the band(s) corresponding to factor H-related proteins 1 and 3 (FHR1-3) were absent, a deletion of FHR1-3 proteins was reported. For FHR1-3 proteins, the immunoblot demonstrates two isoforms (FHR1β and FHR1α). FHR1β is embedded in the same band with the factor H-like protein-1 (FHL-1), but the intensity of FHR1α can be compared to the standards with normal or half-normal levels of FHR1α to indicate homozygous or heterozygous deletion, respectively. Since the FHR1 and FHR3 genes are tightly linked, in most cases, the FHR1 deletion also encompasses FHR3. This observation has been verified by a Multiplex Ligation Dependent Probe Amplification (MLPA) test.

C3 activation and C3 activating factors in patient sera were analyzed using serum mixing tests and immunoblotting. Different mixtures were prepared as follows: (1) 100 μL patient serum and 2 μL 5 M EDTA, (2) 50 μL patient serum, 50 μL donor serum, and 2 μL 5 M EDTA, (3) 50 μL patient serum, 50 μL donor serum, and 2 μL 5 M EDTA, (4) 50 μL patient serum, 50 μL donor serum, and 2 μL PBS and (5) 50 μL patient serum, 50 μL donor serum and 2 μL 1 M MgCl_2_/5 M EGTA. Mixtures 1–2 were incubated for 1 h at 4 °C to stop the reaction and mixtures 3–5 at 1 h at 37 °C to allow the reaction to continue. For mixtures 2 and 3, the mixtures were prepared similarly, but the incubation temperature was different in order to see the baseline and whether any cation-independent C3 conversion (e.g., by microbial proteases) would occur. After incubation, the reactions of mixtures 3–5 were stopped with 2 μL 5 M EDTA. To appropriately diluted samples 1–5, 3 μL of reducing agent and 7.5 μL of buffer were added, then they were incubated for 10 min at 70 °C. Twenty μL portions of prepared samples were added to 4–12% SDS-PAGE gels and ran and transferred onto a filter membrane similar to factor H immunoblotting. After that, non-specific binding sites were blocked in a similar manner as factor H immunoblotting; however, 10 mL was used. To the blocking liquid, 1:10,000 polyclonal rabbit anti-human anti-C3c-antibody (Dako, Glostrup, Denmark) was added and incubated overnight at 4 °C, except for patients 10–17, for whom a polyclonal sheep anti-human C3c-antibody (Bio-rad Laboratories, Solna, Sweden) was used. Similarly, for factor H immunoblotting, membranes were washed with PBS. Next, HRP-conjugated goat anti-rabbit IgG secondary antibody (Dako, Glostrup, Denmark) was added in a dilution of 1:10,000 (into a 5% milk power and PBS with 0.05% Tween 20). For patients aged 10–17, a donkey anti-sheep HRP-conjugated secondary antibody (Jackson ImmunoResearch, Ely, UK) was used. Similar to factor H testing, washing with PBS was carried out for 1 h, and an in-house protocol for enhanced electrochemiluminescence was performed for visualization. Films were then developed at variable exposure times. See the full list of complement analyses performed in Appendix A. The reference values for complement tests were as described for each analysis.

### 2.6. Statistical Analysis

Statistical analyses were performed by a professional biostatistician using R software version 4.0.4 (R Core Team, 2021). Mean values were compared using the t-test of independent samples. As the mean estimator was assumed to be asymptotically distributed regardless of the distribution of the variable itself, and as we wanted to emphasize the difference in distribution, the parametric t-test was chosen. Categorical data analysis was carried out using Fisher’s exact test. In addition, logistic regression was used to predict factors determining the changes in binary dependent variables, and the difference of survival functions between groups was estimated using a Kaplan–Meier curve [40], and the corresponding statistical test used was a log-rank test. The significance level (*p*-value) of all statistical tests was set to 0.05.

## 3. Results

### 3.1. Patient Population

A preliminary data search of 7078 adult biopsies identified 204 patients, out of which 60 (0.8%) fulfilled the inclusion criteria. The relative annual incidence of MPGN varied between 0.4–1.5%. After index biopsy re-evaluation, there were 37 (62%) IC-MPGN patients and 23 (38%) C3G patients (including one DDD patient). Index biopsy was the diagnostic biopsy for 52 (87%) patients, and for 8 (13%) patients, the diagnosis was from an earlier biopsy. Nine (15%) index biopsies were from kidney transplants. Out of the entire study population, 42 (70%) were primary MPGN and C3G cases. No misdiagnosis among TMA patients and no masked immunoglobulins were observed after 24 PIF examinations. However, 4/11 (36%) PIGN patients were reclassified as either C3G or IC-MPGN. Twenty-nine (29/60, 48%) patients attended the outpatient visit, for which detailed complement analyses were performed.

### 3.2. Clinical Characteristics

Baseline characteristics at the time of clinical diagnosis are summarized in Table 1 and Table 2. No statistically significant differences between IC-MPGN and C3G patients were found in the patient or kidney characteristics or the other laboratory variables. Estimated GFR was below normal (≤60 mL/min/1.73 m^2^) in 63% and 73%, and nephrotic range proteinuria was detected in 68% and 43% of the IC-MPGN and C3G patients, respectively. The mean ages were 52 and 54 years, and the proportions of males were 62% and 57% for IC-MPGN and C3G groups, respectively.

### 3.3. Histological Characteristics

No statistically significant differences in the index biopsy histological analyses were found between the two patient groups (Table 3). However, in the C3G group, there seemed to be a higher frequency of mild interstitial fibrosis (48% in C3G and 11% in IC-MPGN, *p* = 0.059). The IF and EM findings between the groups are presented in Appendix A, which showed no differences in EM findings. The IF findings differed between the groups, as the diagnosis of IC-MPGN and C3G is based on these findings. None of the studied morphological variables predicted progressive kidney disease in multivariate analysis (Appendix A). Various LM features possibly representing a different phase of the injury process were observed (Figure 2), and their ratios in IC-MPGN and C3G patients are portrayed in Figure 3. Classical MPGN-type patterns were the most common, but overall, it was observed in only 34% of cases. The various LM features portrayed in Figure 3 did not predict dialysis (*p* = 0.189), transplantation (*p* = 0.814), or death (*p* = 0.996) in either of the groups for the entire study population, or for the primary MPGN and C3G cases (*p* = 0.451, *p* = 0.763, and *p* = 0.973, respectively).

### 3.4. Baseline Complement and Paraprotein Findings

Complement variables at baseline were available for only a subset of patients. Serum C3 was decreased in two (20%, *n* = 10) C3G patients, but in none of the IC-MPGN patients (*n* = 7). Serum C4 was decreased in one (13%) C3G and two (18%) IC-MPGN patients. Functional complement analysis showed that alternative pathway activity was lower than the reference value in two (29%) C3G and one (13%) IC-MPGN patients. Classical pathway activity was decreased in two (25%) and six (50%) patients, and lectin pathway activity in one (14%) and one (13%) patient, respectively. Four (50%) C3G patients and three (25%) IC-MPGN had positive C3 nephritic factor, but none had factor H-antibody positivity. Serum paraprotein was detected in five (33%) C3G and eight (28%) IC-MPGN patients and urine paraprotein in four (27%) and one (4%) patients, respectively.

### 3.5. Complement and Paraprotein Findings at Follow-Up

Complement and paraprotein characteristics at the study follow-up visit are summarized in Table 4. Insignificantly, the alternative and classical pathway activities were lower in the C3G than in the IC-MPGN group. Lectin pathway activity was most often decreased below the reference value in the whole study population (34%), and more frequently in IC-MPGN patients (44%) than in C3G patients (21%). A heterozygous deletion of FHR1-3 was detected in 30% of all the patients. Factor H-antibody positivity was seen in 6% of all the patients. A C3 activation test showed an ability to activate the classical or the alternative pathway by serum factors from four IC-MPGN patients (12% and 12%, respectively).

### 3.6. Treatment, Kidney, and Patient Survival

Treatments at baseline and during follow-up did not differ between the patient groups (Table 5). Follow-up time for the whole study population was 7.3 (range 0.08–38) years, during which 11 (49%) C3G and 16 (43%) IC-MPGN patients developed progressive kidney disease. Blood pressure medication was used on 36 IC-MPGN patients, on 20 C3G patients at baseline, and on 37 and 23 patients during follow-up, respectively. Eculizumab was used only for one C3G patient during the follow-up. Time from diagnostic biopsy to the start of kidney replacement therapy, to kidney transplantation, or to death was not significantly different between the groups for the entire study population (Figure 4a, Figure 5a and Figure 6a) or for the primary MPGN and C3G cases (Figure 4b, Figure 5b and Figure 6b). Twelve (32%) IC-MPGN patients and 5 (22%) C3G patients died during the study follow-up. At the last study follow-up, twelve (20%) patients out of the entire study population had received a kidney transplant.

Multivariate analyses for histological, clinical, and laboratory baseline variables contributing to the progression showed that no histological feature was of prognostic value. Baseline eGFR was associated with disease progression in the entire study population (OR 1.0, 95% CI 0.9–1.0, *p* = 0.040), but not in the C3G or IC-MPGN subgroups. Serum albumin level was associated with progression in C3G (OR 1.7, CI 1.1–2.8, *p* = 0.03), but not in IC-MPGN or the whole study population (Appendix A).

## 4. Discussion

Kidney diseases IC-MPGN and C3G have been reported to have similar features. As the presumed pathogenetic processes behind both diseases are diverse and partially overlapping it has remained uncertain whether they could be classified as separate syndromes. Our study shows that the current diagnostic means do not separate the two disease complexes, and their prognoses do not considerably differ either. Because of the multiple causative factors, more precise etiology-based definitions of disease subcategories are needed. As an example, studies on the mechanisms of paraprotein-related disease forms require further attention.

Our study consisting of 60 IC-MPGN and C3G patients represents the first analysis of adult MPGN patients in the genetically unique Finnish population. A Korean multicenter retrospective study discovered that the incidence of MPGN was 2.3% [41], which is somewhat higher than what we observed (0.4–1.5%). The multiple predisposing and etiological factors that contribute to the development of MPGN include infections, autoimmunity, abnormalities in complement activation or regulation, paraproteins, and genetic factors. Several drugs that target the complement system are under investigation for C3G and IC-MPGN, but clinical trials will be challenging and heavily influenced by the heterogeneity of the diseases [42].

In our study, 28% and 22% of IC-MPGN and 33% and 40% of C3G patients had serum paraprotein at baseline and at follow-up visits, respectively. In another study, 41% of MPGN patients had monoclonal gammopathy [43], while the proportion was 20% in a study of 60 patients with proliferative glomerulonephritis, with an MPGN pattern in only 48% [44]. A study including 14 DDD patients demonstrated that 71% of these patients had MGUS [18], which is significantly higher than in our study. The mechanisms behind complement-mediated kidney damage are unknown in most cases. For example, a nephritogenic lambda light chain was found to act as a mini-autoantibody against complement factor H in a patient with DDD/MPGN already many years ago [45]. Current evidence suggests that both polyclonal and monoclonal immunoglobulins seem to be involved in immunoglobulin-associated C3G [46]. Any disease causing chronic antigenemia can lead to immune-complex formations and their deposition in the glomeruli. The immune complexes trigger the activation of the classical pathway of complement, leading to the deposition of complement factors. Immunofluorescence typically shows immunoglobulins and complement, and the subsequent morphology of IC-MPGN. Paraproteins are known to be able to cause IC-MPGN and also C3G in cases where paraprotein acts as an autoantibody against complement protein [4]. The underlying mechanisms of complement activation and related pathophysiology are not yet fully understood. Some paraproteins may act as autoantibodies against complement proteins. In these cases, the targets could be factor H, factor B, C3bBb (C3 nephritic factor), or C3b. Other possibilities could be complexes of the paraproteins themselves or immune complexes formed by the paraprotein binding to a glomerular antigen, or to an antigen “planted” within the glomeruli, e.g., as a consequence of infection or tissue trauma.

### 4.1. Clinical Course, Survival Probability, and Complement Analyses

The clinical course in our patients was severe. This is indicated by the fact that eGFR at baseline was abnormal in the majority of the patients in our study. A progressive disease was observed in almost half of the patients, which is in accordance with some previous studies [19,41]. The two subgroups seemed to resemble each other, as we did not observe significant differences in the clinical course or the survival probability. We chose to include both primary and secondary MPGN cases in our study in order to present a comprehensive patient population of a type that clinicians encounter in their everyday work, in spite of primary and secondary MPGN patients requiring different treatments. We did not, however, observe significant differences in the prognosis of dialysis, transplantation, or death including only primary MPGN and C3G patients in statistical tests. These points suggest that the current subdivision of MPGN does not add substantial value to the assessment of the disease course or prognosis. Similar results were achieved by a clustering analysis of 173 native biopsy patients, in which primary IC-MPGN and C3G were divided into four categories based on multiple clinical and histological features including results from the complement analyses. It seemed that belonging to the cluster consisting of patients with normal levels of blood C3, sC5b-9, and intensive renal biopsy C3 staining was an independent determinant of end-stage renal disease. These were linked to certain histological features (crescents and the number of sclerosed glomeruli) and nephrotic range proteinuria [47]. Collectively, the observations suggest that overactivation of the alternative pathway leads to endothelial damage. Addressing the importance of complement measurements, it was shown in another study that normal C3/high sC5b-9 levels, or low C3/normal sC5b-9 levels, remained independently associated with a worse kidney prognosis in C3G with adult onset of the disease [48].

Many new drugs targeting various sites in the complement cascade are also in clinical trials for MPGN patients [11]. To understand the best treatment for each patient, it is important to study the complement abnormalities in detail. Additionally, it would be important to know the dynamics of complement derangements in the course of the disease. The functions and persistence of autoantibodies should also be clarified. Information about the background and mechanism of the disease is important because a similar medication is unlikely to be suitable for every patient and optimal treatment may even change during the course of the illness. In our study, a form of complement autoantibody was detected in approximately 30% of all patients. Of these, C3Nef was found in 18% of C3G but in none of the IC-MPGN patients at the follow-up visit. Unfortunately, complement variables were not uniformly tested at the time of diagnosis, which prevented proper analyses over the course of the disease. Indeed, this might explain the low rate of hypocomplementemia at diagnosis detected in our study. It is well-known that hypocomplementemia can also fluctuate and the rate of hypocomplementemia at the last follow-up visit could represent the result of various treatments and a more stable phase of the disease. Regardless, out of the tested C3G and IC-MPGN patients at diagnosis, complement autoantibody positivity was found in 50% and 25% of the patients, respectively. These figures are similar in C3G but substantially smaller in IC-MPGN patients than described elsewhere [49]. Factor B and C3b autoantibody positivity in our study was detected in equal numbers in 15% and 6% of C3G and IC-MPGN patients, respectively. Factor B antibodies have been described in 14% of C3G patients in a retrospective study [50], and in 8.5% in a study that investigated all MPGN patients (primary and secondary IC-MPGN and C3G patients) retrospectively [51]. The latter study revealed anti-C3b positivity in 5.7% of patients and a minority were double positive for C3b and factor B (4%) [51].

### 4.2. Histological and Clinical Features Predicting Prognosis

Histological light microscopy variables did not significantly differ between the patient groups, but there was a slightly increased frequency of mild interstitial fibrosis in the C3G group, as opposed to the IC-MPGN group. It is unclear whether this reflects the timing of the biopsy as opposed to the course of the disease. Nonetheless, based on the multivariate analysis of histological factors, the amount of fibrosis did not seem to affect the prognosis of the disease. This may merely reflect the low number of patients, as previous studies have concluded that the number of sclerosed glomeruli, crescents, tubular atrophy, and interstitial fibrosis predicts progression in IC-MPGN and C3G patients [47,52]. It is of notice that LM morphology was variable, and in addition to the classical MPGN-type pattern, other morphologies were evident both in IC-MPGN and C3G. It is possible that the different LM morphologies observed in our study could harbor different prognoses, but as the study population is small, we are unable to test it with all the five different morphological subgroups observed in our study. However, for the entire study population and for the primary and C3G cases, the prognosis of dialysis, transplantation, and death did not show significant differences.

Estimated glomerular filtration rate (eGFR) at baseline, 24 h proteinuria, and treatment with immunosuppression were the main determinants of kidney failure in a model with only clinical variables in a study of 111 C3G patients [52]. Kidney function and the amount of proteinuria are generic determinants of poor kidney prognosis in virtually all proteinuric kidney diseases, but only baseline eGFR was a significant prognostic determinant in our study.

### 4.3. Strenghts and Limitations of the Study

Our study was conducted within one hospital district, which ensured that relevant clinical data and the majority of biopsies were available for re-evaluation. In addition, approximately half of the patients attended the research outpatient follow-up visit, allowing further complement testing.

While this study was informative, there were some limitations. As we chose to include both primary and secondary MPGN patients in our study in order to present a real-life situation and in order to maintain an adequate study population for statistical purposes, the patient populations with primary and secondary MPGN cases remain heterogenous, limiting the analyses of treatment protocols and prognosis. Moreover, MPGN and C3G are rare kidney disorders, which causes the number of patients included in this study to be limited even when including both primary and secondary cases. This limits the ability of statistical tests to show significant differences. Moreover, as the study group consisted of IC-MPGN and C3GN patients, only one DDD patient was enrolled, preventing the study of differences between DDD and C3GN. The fact that pediatric patients were excluded may have influenced this finding, as DDD is more commonly seen in children. Additionally, diagnostic biopsies from some patients were taken before the research period and were not available for re-evaluation. Moreover, we were unable to have results for detailed complement testing at baseline due to the retrospective nature of the study. At a follow-up visit, the pattern of functional complement testing did not include terminal pathway activation, C4 nephritic factor (C4Nef), or C5 nephritic factor (C5Nef), which could be implemented in the armamentarium of complement tests in these patients. Moreover, approximately half of the patients did not attend the research outpatient visit limiting the comprehensive laboratory, immunologic, and genetic testing performed. In addition, further investigations including genetic testing and MLPA analyses of the FH/FHR gene region are warranted. The fact that the patients were enrolled in only one hospital district may limit the generalizability of these findings. The low number of patients limited the use of thorough multivariate analyses.

## 5. Conclusions

The current division of patients into IC-MPGN and C3G diagnostic groups did not reveal substantial differences in the clinical course, nor in the overall or renal survival. This might suggest that subdividing these patients into currently widely used groups may be unnecessary. Perhaps clustering these patients and taking multiple factors beyond histology into consideration could help with decisions on the prognosis and optimal treatments of MPGN patients. Histological features in MPGN can manifest as different types of morphology depending on the timing of the biopsy and the activity or chronicity of the disease, and it does not always manifest as a classical MPGN-type pattern. Further detailed analyses of individual patients are important in delineating the precise mechanisms of the underlying diseases.

## Figures and Tables

**Figure 1 cells-12-00712-f001:**
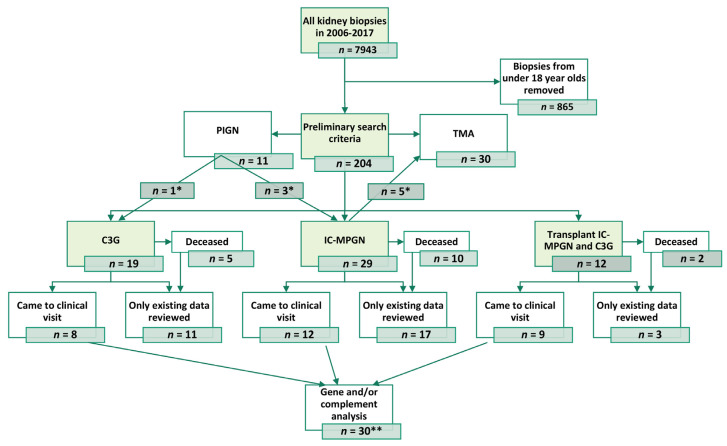
Study flow of the research patients. * After index biopsy re-evaluation, patients were reassigned to an accurate diagnostic category, ** For one C3G patient complement analysis sample was acquired before research visit and included in complement analysis (total *n* = 30). PIGN = postinfectious glomerulonephritis, TMA = thrombotic microangiopathy, C3G = C3 glomerulopathy, IC-MPGN = immune-complex-mediated glomerulonephritis.

**Figure 2 cells-12-00712-f002:**
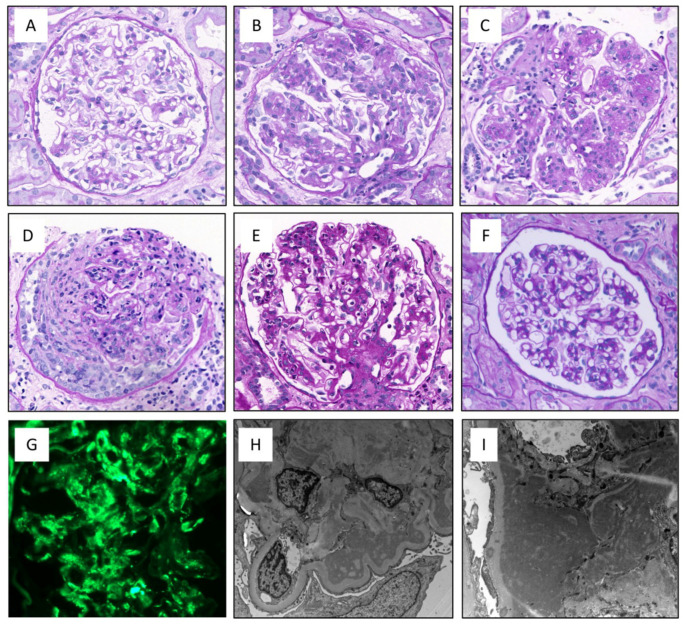
Different morphological features of the index biopsies of IC-MPGN and C3G patients may indicate different activity of the glomerulonephritis (GN) and/or the time point of injury. (**A**) Minimal to mild mesangial expansion, (**B**) mesangioproliferative injury, (**C**) classical membranoproliferative injury with lobular/nodular proliferation and double contours in basement membranes, (**D**) crescentic injury, (**E**) exudative GN that can be difficult to distinguish from postinfectious GN in the early phase. However, it usually resolves within 3–6 months, whereas C3G or IC-MPGN do not, (**F**) Normal glomerulus for reference, (**G**) IF showing coarse granular C3 positivity in mesangial areas and basement membranes, (**H**,**I**) Electron-dense deposits on subendothelial and mesangial spaces. (**A**–**F**) Periodic Acid-Schiff staining, (**G**) frozen section C3 immunofluorescence staining, (**H**,**I**) electron micrograph.

**Figure 3 cells-12-00712-f003:**
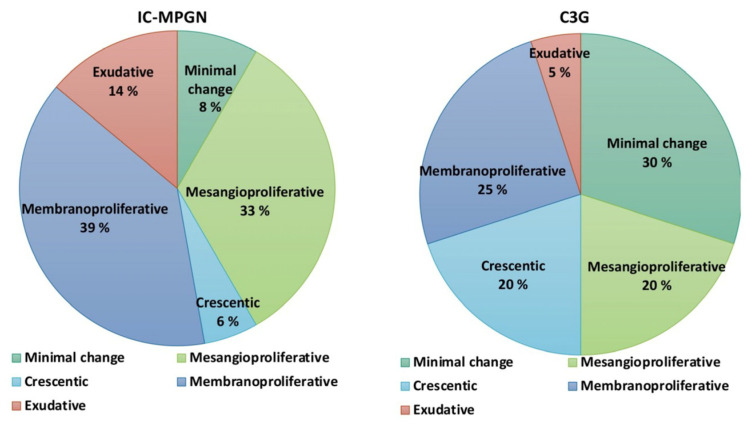
Different morphological features were observed on light microscopy (LM) in IC-MPGN and C3G patients. Differences in subcategory division between the patient groups were not significant (*p* > 0.05 for comparisons).

**Figure 4 cells-12-00712-f004:**
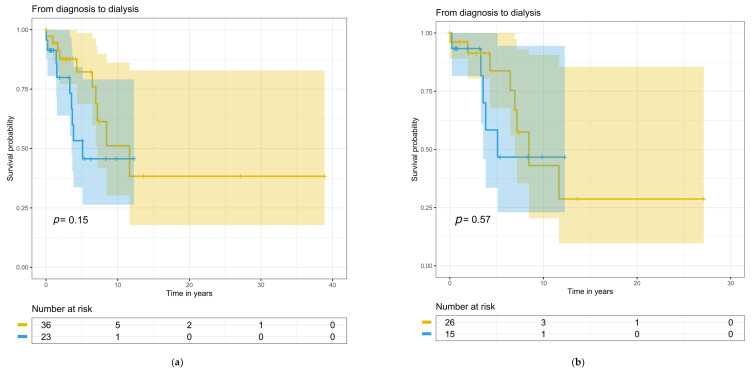
Longitudinal changes from diagnosis to dialysis with numbers at risk: (**a**) for the entire study population; (**b**) for primary MPGN and C3G cases. The yellow line indicates IC-MPGN and the blue line indicates C3G. Colored areas indicate confidence interval and small vertical lines indicate the end of follow-up.

**Figure 5 cells-12-00712-f005:**
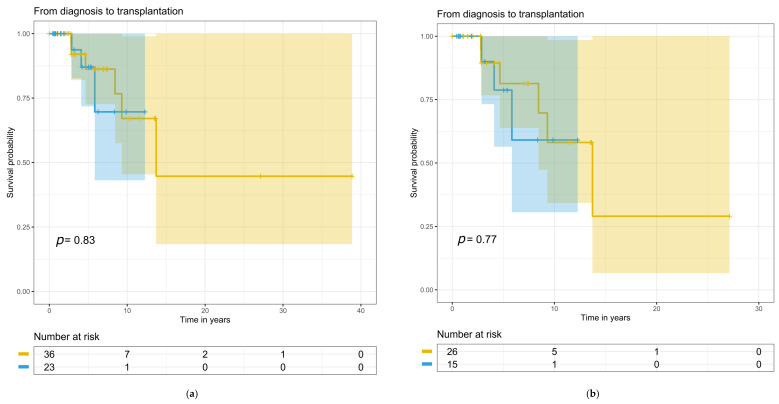
Longitudinal changes from diagnosis to transplantation with numbers at risk: (**a**) for the entire study population; (**b**) for the primary MPGN and C3G cases. The yellow line indicates IC-MPGN and the blue line indicates C3G. Colored areas indicate confidence interval and small vertical lines indicate the end of follow-up.

**Figure 6 cells-12-00712-f006:**
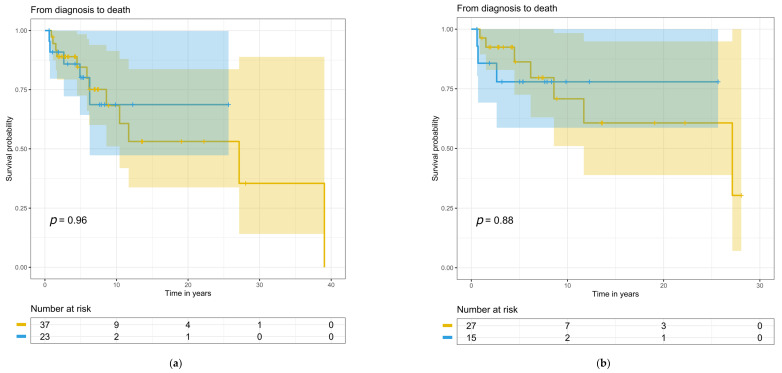
Longitudinal changes from diagnosis to death with numbers at risk: (**a**) for the entire study population; (**b**) for primary MPGN and C3G cases. The yellow line indicates IC-MPGN, and the blue line C3G. Colored areas indicate confidence interval and small vertical lines indicate the end of follow-up.

**Table 1 cells-12-00712-t001:** Patient characteristics at baseline. Values are expressed as means (range or SD) or numbers (percentage) of patients. Reference values and units are given where appropriate. The number of patients indicates the maximum number of patients available with the information. Unless otherwise indicated, the parameter values were available for all patients. The differences between the groups were not significant (*p* > 0.05 for all comparisons).

Baseline Variable	IC-MPGN, *n* = 37	C3G, *n* = 23
Age, years (range)	52 (5–78)	54 (16–79)
Male sex, *n* (%)	23 (62)	13 (57)
BMI (kg/m^2^) (SD)	28.8 (6.3), *n* = 34	26 (4.3), *n* = 22
Smoking		
Current, *n* (%)	9 (27), *n* = 34	7 (32), *n* = 22
Former, *n* (%)	14 (41), *n* = 34	11 (50), *n* = 22
Diabetes, *n* (%)	2 (6), *n* = 34	4 (17)
Hypertension, *n* (%) *	36 (97)	23 (100)
Rheumatic disease, *n* (%)	2 (6), *n* = 34	4 (17)
Chronic infection, *n* (%)	7 (21), *n* = 34	3 (13)
Malignancy, *n* (%)	2 (6), *n* = 34	4 (17)
Plasma cell dyscrasia, *n* (%) **	8 (24), *n* = 34	6 (26)
Cardiovascular disease, *n* (%)	10 (29), *n* = 34	7 (30)
Duration of renal findings before diagnosis, years, (range)	1.3 (0–9), *n* = 27	1.4 (0–7)
Macroscopic hematuria, *n* (%)	4 (13), *n* = 31	6 (27), *n* = 22
Diagnostic biopsy		
Transplant, *n* (%)	1 (3)	2 (9)
Native kidney, *n* (%)	36 (97)	21 (91)
Diagnostic biopsy taken before index biopsy, *n* (%)	7 (19)	1 (4)

SD = standard deviation, BMI = body mass index. The upper limit of normal for hypertension is 135/85 mmHg. * Hypertension is defined as systolic and/or diastolic blood pressure that is over the limit and/or the use of antihypertensive medication; ** myeloma is excluded.

**Table 2 cells-12-00712-t002:** Renal function and laboratory variables at baseline. Values are expressed as means (SD) or numbers (percentage) of patients. Reference values and units are given where appropriate. Number of patients signifies the maximum number of patients available with the information. If variable values were not available for all patients, it is expressed in the table. None of the differences between the groups were significant (*p* > 0.05 for all comparisons).

Baseline Variable	IC-MPGN, *n* = 35	C3G, *n* = 23
eGFR (SD) *	56 (32)	49 (27), *n* = 22
eGFR decreased, *n* (%)	22 (63)	16 (73), *n* = 22
S-creat, µmol/L (SD)	155 (99)	166 (121)
Urine dipstick proteinuria positive, *n* (%)	33 (97), *n* = 34	21 (100), *n* = 21
Urine alb/creat, mg/mmol (SD)	192.6 (160.6), *n* = 15	158.8 (171.5), *n* = 9
Urine protein excretion, g/24 h (SD)	5.3 (4.2), *n* = 34	3.6 (3.3), *n* = 21
Nephrotic proteinuria (<3 g/24 h), *n* (%)	23 (68), *n* = 34	9 (43), *n* = 21
Microscopic hematuria, *n* (%)	28 (90), *n* = 31	20 (100), *n* = 20
Alb, g/L (SD)	27 (6)	25 (6)
Hb, g/L (SD)	116 (18)	116 (19)
CRP, mg/L (SD)	13 (20)	37 (81)
LDL, mmol/L (SD)	3.5 (1.5)	3.3. (1.7)
ENA-Ab positivity, *n* (%)	0 (0)	1 (14)

SD = standard deviation, Creat = creatinine (upper limit of normal ≤ 100 for male and ≤ 90 for female), Urine alb/creat = urine albumin/creatinine (upper limit of normal 2.5 mg/mmol for male, 3.5 mg/mmol for female), Alb = albumin (36–45 g/L), Hb = hemoglobin (134–167 for male, 117–155 for female), CRP = C-reactive protein (upper limit of normal 4 mg/L), LDL = low-density lipoprotein (upper limit of normal 3 mmol/L), ENA-Ab = extractable nuclear antigen antibody. * eGFR = estimated glomerular filtration rate, calculated according to the Chronic Kidney Disease Epidemiology Collaboration (CKD-EPI) equation (EGFR lower limit of normal ≥ 60 mL/min/1.73 m).

**Table 3 cells-12-00712-t003:** Histological parameters of index biopsies. Values are expressed as means (SD) or numbers (percentage) of patients. The number of patients signifies the maximum number of patients with the information available.

Variable	IC-MPGN, *n* = 37	C3G, *n* = 21 *	*p*-Value
Glomerular changes			
% of sclerotic glomeruli	17 (47), *n* = 36	11 (52)	1.000
Biopsies containing crescents, *n* (%)	5 (13.5)	4 (19)	0.710
Mesangial matrix expansion			
none, *n* (%)	10 (27)	6 (29)	1.000
mild, *n* (%)	8 (22)	6 (29)	0.761
moderate, *n* (%)	6 (16)	2 (10)	0.703
strong, *n* (%)	13 (35)	7 (33)	1.000
Lobulated pattern of glomeruli, n (%)	20 (54)	9 (43)	0.811
Doubled GBM			
None, *n* (%) Mild, *n* (%) Moderate, *n* (%) Strong, *n* (%)	3 (8) 7 (19) 3 (8) 24 (65)	4 (19) 5 (24) 2 (10) 10 (48)	0.415 0.751 1.000 0.648
Tubulointerstitial changes			
Total interstitial inflammation			
None, *n* (%) Mild, *n* (%) Moderate, *n* (%) Strong, *n* (%)	27 (73) 4 (11) 4 (11) 2 (5)	10 (48) 8 (38) 2 (10) 1 (5)	0.379 0.062 1.000 1.000
Interstitial fibrosis			
None, *n* (%) Mild, *n* (%) Moderate, *n* (%) Strong, *n* (%)	29 (78) 4 (11) 2 (5) 2 (5)	10 (48) 9 (43) 2 (10) 0 (0)	0.375 0.059 0.623 0.537
Tubular atrophy			
None, *n* (%) Mild, *n* (%) Moderate, *n* (%) Strong, *n* (%)	24 (65) 11 (30) 0 (0) 2 (5)	10 (48) 10 (48) 0 (0) 1 (5)	0.648 0.437 1.000 1.000
Arteriolar sclerosis			
None, *n* (%) Mild, *n* (%) Moderate, *n* (%) Strong, *n* (%)	17 (49), *n* = 35 12 (34), *n* = 35 6 (17), *n* = 35 0 (0), *n* = 35	11 (52) 8 (38) 2 (10) 0 (0)	1.000 1.000 0.7 1.000

* Two C3G patients had inadequate material for light microscopy (LM) but were adequate for immunofluorescence (IF) and electron microscopy (EM) to conclude the diagnosis. GBM = glomerular basement membrane.

**Table 4 cells-12-00712-t004:** Complement and paraprotein characteristics during the last outpatient visit. Values are expressed as means (SD) or numbers (percentage) of patients. The number of patients signifies the maximum number of patients with the information. Unless otherwise indicated, the variable numbers were available for all patients. None of the differences between the groups were significant (*p* > 0.05 for all comparisons).

Variable	IC-MPGN, *n* = 23	C3G, *n* = 17
Complement proteins		
S-C3, g/L (SD) Decreased (lower limit of normal 0.5 g/L), *n* (%)	1.0 (0.2), *n* = 21 1 (5)	0.9 (0.3) 1 (7)
S-C4, g/L (SD) Decreased C4 (lower limit of normal >0.12), *n* (%)	0.2 (0.1), *n* = 20 2 (10)	0.2 (0.1) 3 (21)
FHR1-3 heterozygous deletion, *n* (%)	5 (29), *n* = 17	4 (31), *n* = 13
Functional complement analysis		
S-CH100AI, % (SD) Decreased, *n* (%)	93 (31), *n* = 18 1 (6)	75 (33), *n* = 14 2 (14)
S-CH100CI, % (SD) Decreased, *n* (%)	95 (25), *n* = 18 2 (11)	77 (34), *n* = 14 4 (29)
S-CH100L, % (SD) Decreased, *n* (%)	50 (53), *n* = 18 8 (44)	58 (42), *n* = 14 3 (21)
Complement autoantibodies		
C3Nef positivity, n (%)	0 (0), *n* = 22	2 (18), *n* = 11
Factor H-antibody positivity, *n* (%) *	1 (6), *n* = 18	1 (8), *n* = 13
Factor B antibody positivity, *n* (%)	1 (6), *n* = 17	2 (15), *n* = 13
C3b-antibody positivity, *n* (%)	1 (6), *n* = 17	2 (15), *n* = 13
C3 activating factors		
classical pathway activator, *n* (%) alternative pathway activator, *n* (%)	2 (12), *n* = 17 2 (12), *n* = 17	0 (0), *n* = 13 0 (0), *n* = 13
**Paraproteins**		
Free kappa light chain in serum, mg/L (SD)	66.9 (77.2)	79 (80.9), *n* = 14
Free lambda light chain in serum, mg/L (SD)	37.4 (31.9)	64.5 (53), *n* = 14
Kappa/lambda light chain ratio (SD)	1.6 (1)	1.2 (0.4), *n* = 15
Serum paraprotein, *n* (%)	5 (22)	6 (40), *n* = 15
Urine paraprotein, *n* (%)	0 (0), *n* = 16	3 (25), *n* = 12

C3 = complement C3 (0.5–1.5 g/L), C4 = complement C4 (0.12–0.42 g/L), FHR1-3 = factor H-related-protein 1–3 (deletion determined by immunoblotting), CH100Al = activity of the alternative pathway of complement (<39%), CH100Cl = activity of the classical pathway of complement (>74%), CH100L = activity of the lectin pathway of complement (>10%), C3Nef = Complement C3 nephritic factor, free kappa light chains (6.9–25.6 mg/L), free lambda light chains (8.6–26.5 mg/L), kappa/lambda light chain ratio (0.52–1.40). * Weakly positive (+/−) results are also considered positive.

**Table 5 cells-12-00712-t005:** Treatment, including medications at baseline and during follow-up as well as data on kidney replacement therapy and follow-up. Values are expressed as means (range or SD) or numbers (percentage) of patients. Number of patients signifies the maximum number of patients with the information. Unless otherwise indicated, the variable numbers were available for all patients. Immunosuppressive medication denotes treatment aimed at kidney disease (not due to kidney transplantation). Differences between the groups were not significant (*p* > 0.05 for all comparisons).

Variable	IC-MPGN, *n* = 37	C3G, *n* = 23
**Immunosuppressive medication at baseline**		
Corticosteroids, *n* (%) Mycophenolate mofetil, *n* (%) Other, *n* (%) *	3 (9), *n* = 34 0 (0), *n* = 33 1 (3)	6 (26) 1 (5) 1 (4)
**Immunosuppressive medication during follow-up**		
Corticosteroids, *n* (%) Mycophenolate mofetil, *n* (%) Other, *n* (%) *	17 (46) 5 (14) 8 (22)	13 (57) 6 (26) 7 (30)
**Follow-up**		
Dialysis during follow-up, *n* (%)	8 (22)	7 (30)
Kidney transplantation during follow-up, *n* (%)	8 (22)	4 (17)
Progressive disease, *n* (%)	16 (43)	11 (48)
More than one kidney transplant, *n* (%)	3 (8)	1 (4)
Follow-up time from diagnostic biopsy, years (range)	8.1 (0.9–39.1)	5.9 (0.6–25.7)
Time from diagnostic biopsy to start of 1st dialysis, months (range)	59 (0.2–140), *n* = 10	30 (0.2–61), *n* = 9
Time from diagnostic biopsy to 1st transplantation, months (range)	83 (33–165), *n* = 6	51 (34–70), *n* = 3

* Cyclosporin A, tacrolimus, azathioprine, rituximab, or cyclophosphamide.

## Data Availability

The data that support the findings of this study are available from the corresponding author, M.K., upon reasonable request. The data that support the findings of this study are not publicly available due to ethical reasons and due to privacy reasons of research participants. Data availability can be requested from the corresponding author, M.K., upon reasonable request.

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
