# Peer review of "Diagnostic and Prognostic Comparison of Immune-Complex-Mediated Membranoproliferative Glomerulonephritis and C3 Glomerulopathy"

_cells, 2023, doi:10.3390/cells12050712_

Round 1

Reviewer 1 Report

1. For the patients’ description, native biopsy must be explained in this section

2. In part 2.2. the name of commercial kit and its company must be mentioned for autoantibody profile
assay (8 ENA antigens). Also, a brief description of the method is necessary.

3. In part 2.5. for assessment of C3 activation, the authors denoted using 5 different mixtures of patients serum for this purpose. But there is no difference between mixture 2 and 3! Please check it.

4. In table 2, urine albcrea must be Alb/Creat

5. Supplementary table 2 is confusing. Please check and revise it

6. In table 3, it seems to be higher frequency of mild interstitial fibrosis in the C3G group of the patient(P=0.059, marginally significant!). But, it has not been discussed in the discussion part.

7. Based on reporting heterozygous deletion evaluation in the results (part 3.5), I think the methods lack a short explanation for this assessment (How they perform this test?)

8. Table 5: the number of transplant patients is 9 inside the text and inside the flowchart of the study but, is 12 inside the table (8 in IC-MPGN vs. 4 in C3G group)!!. Please check it.

Point 9. In the third paragraph, the possible mechanisms of involvement of paraproteins in immune mediated-MPGN needs to be more discussed in relation to complement factors.

Reviewer 2 Report

This is a retrospective cohort study of patients with immune-complex mediated membranoproliferative glomerulonephritis (IC-MPGN) and C3 glomerulopathy (C3G) to clarify the differences between the two diseases. Authors demonstrated that there were no clinical or histological differences between the two diseases, and the prognosis was similar.

The presented study was well performed, and the manuscript is described in a reasonable manner. Due to the rarity of this disease, this study included transplant patients and patients with both primary and secondary MPGN to increase the number of cases. However, transplant patients and patients with MPGN require different immunosuppressive regimens, and patients with primary and secondary MPGN require completely different treatments, so I think that the subjects in this study are inappropriate to evaluate the prognosis. The conditions of subjects had better to be limited to patients with primary MPGN due to the small number of cases.

In addition, I have several concerns:

1.     I think it's better to remove reference values in Tables. On the other hand, I think it would be better to clarify the definition of variables. For example, in the case of Table 4, it would be easier to understand if the ref value of serum C3 concentration was deleted and the definition of low serum C3 concentration (i.e. < 0.5 g/L) was described.

2.     In Table 3, please provide the immunofluorescence and electron microscopy findings in addition to the light microscopy findings.

3.     I think the rate of hypocomplementemia is low, what do authors think?

4.     In Table 4, I think it makes little sense to assess complement and paraprotein levels at the last visit, post-treatment. Instead, I think it would be better to evaluate their values at initial presentation and renal biopsy.

5.     In Figure 4, numbers of risk should be added.

6.     Please also provide the prognosis by morphological features.

Reviewer 3 Report

In this manuscript, Kovala and colleagues aimed to delineate the differences between the immune-complex mediated glomerulonephritis (IC-MPGN) and C3 glomerulopathy (C3G) types of membranoproliferative glomerulonephritis (MPGN).

From a total of almost 8000 kidney biopsies, they excluded biopsies of post-infectious glomerulonephritis and thrombotic microantiopathy and included at total of 30 biopsies with a typical diagnosis of C3G, IC-MPGN, and transplant IC-MPGN or C3G. 

The authors then compared kidney histology, complement analysis and clinical data of the patients included. Patient characteristics and renal function showed no significant difference between the groups. Histology of the renal biopsy also showed no significant difference between the groups. Features of light microscopy potentially showed different stages of disease progression. Baseline complement C3 and C4 exhibited differences between IC-MPGN and C3G. Classical and alternative C3 activity was different as well. 

Taken together the study confirmed the current view that IC-MPGN and C3G form a disease spectrum of MPGN with shared pathophysiology, genetic aspects and causes of disease.

Author Response

Thank you for your excellent comments. We aimed to present Finnish MPGN and C3G patients in a detailed perspective in order to gain more insight into the disease spectrum of MPGN. We hope you enjoyed our manuscript.

Reviewer 4 Report

In the present manuscript, Kovala and colleagues compared histological and laboratory features as well as clinical outcomes of patients with biopsy-proven diagnosis of IC-MPGN and C3G referred to a Finnish centre.

The following drawbacks are for the Author’s consideration:

1.     The present analysis included adult patients who received a biopsy-confirmed diagnosis of MPGN between 2006 and 2017, and were followed up to the end of 2019. It is unclear why there is a large gap in time between data collection (until 2019) and the submission of the present manuscript. Actually, an updated retrieval of clinical information from the patient population under investigation would be expected to prolong the mean follow-up and provide more contemporary information.

2.     In the Methods section it was claimed that mean values were compared using t-test (line 239). However, some of the patient characteristics under investigation (e.g., age, follow-up from diagnosis) appear to have a skewed distribution. Such parameters should be reported as median (interquartile range), instead of mean (range), and compared through non-parametric tests (e.g., Wilcoxon signed rank test).

3.     According to the information reported throughout the manuscript, adult patients who received a biopsy-confirmed diagnosis of IC-MPGN or C3G during 2006-2017 were included in the analyses (lines 91-93). Instead, based on the data in Table 1 it seems that pediatric patients were also recruited, with the youngest IC-MPGN and C3G patients being 5 and 16 years old at time of diagnosis, respectively. The inclusion criteria of the study should be clearly outlined.

4.     Regarding the comparison of long-term clinical outcomes (i.e., need for dialysis, kidney transplantation and death) between patients with IC-MPGN and C3G, I have the follow concerns: i) the number of patients included in these analyses was too small (i.e., 37 with IC-MPGN and 23 with C3G) to detect statistically significant differences in the clinical outcomes under investigation. This study limitation should be acknowledged in the Discussion; ii) since mean follow-up from diagnosis was significantly longer in patients with IC-MPGN than in those with C3G, at the very least comparisons should have been performed at the same time point from diagnosis in the two groups; iii) according to the information in Table 5, the longest follow-up from diagnosis in the C3G group was 12.5 years, but it seems to exceed 25 years based on data in Figure 4C; iv) based on data in Figure 4C it seems that all patients with IC-MPGN died during the follow-up. For the sake of clarity, the number of patients who died throughout the observation period in the two study groups should be specified. All the aforementioned issues should be addressed.

5.     In the Discussion the Authors commented on the decline in the number of patients positive for C3Nefs from the time of diagnosis to the last follow-up (lines 457-458). However, this comparison does not appear to be appropriate due to the limited number of patients who had complement workup at diagnosis. I would suggest deleting the above sentence.

Minor points:

-       Estimated glomerular filtration rate at diagnosis was argued to be ≤ 60 mL/min/1.73 m2 at diagnosis in 73% and 63% of patients with IC-MPGN and C3G, respectively, according to the information in the core paper (lines 263-264), but the opposite was the case based upon data in Table 2. Moreover, the number of patients with positive dipstick proteinuria and microscopic hematuria in the C3G group do not correspond with the reported proportions (Table 2). These inconsistencies should be explained or corrected.

-       Figure 4 showed Kaplan-Meier curves for progression to dialysis, kidney transplantation and death in patients with IC-MPGN and C3G. The number of patients at risk at each time point should be outlined.

-       The number of patients who received a kidney transplant during the follow-up appeared to be nine based on the information in the main manuscript (line 370), but 12 according to data in Table 5. Please clarify.

Round 2

Reviewer 2 Report

This is a retrospective cohort study of patients with immune-complex mediated membranoproliferative glomerulonephritis (IC-MPGN) and C3 glomerulopathy (C3G) to clarify the differences between the two diseases. Authors demonstrated that there were no clinical or histological differences between the two diseases, and the prognosis was similar.

The presented study was well performed and the revised manuscript is described in a reasonable manner. Authors had responded to my all comments. However, just because there is no statistical difference between the two groups, it is unacceptable to me to assess in combination the prognosis of primary and secondary glomerulonephritis treated in completely different methods. In addition, I am sorry that my previous question had not been appropriate, but it is presumed that the prognosis differs between minimal change and crescentic injury types, so please also present the prognosis by tissue morphology for each IC-MPGN and C3G.

Reviewer 4 Report

As I have requested, in the Kaplan-Meier curves for progression to dialysis, kidney transplantation and death, the number of patients at risk at each time point has been added. In doing this, it appears that all patients in the two study groups (i.e., immune-complex-mediated glomerulonephritis and C3 glomerulopathy) have died or were lost to follow-up during the study period. Nevertheless, the Authors have highlighted as a strength of the study the notion that half of the patients attended the research outpatient follow-up visit for complement testing. Thus, it can be inferred that such follow-up visit was not a recent. This issue should be acknowledged as a limitation, rather than a strength, of the study.
